# Modification of PLA-Based Films by Grafting or Coating

**DOI:** 10.3390/jfb11020030

**Published:** 2020-05-07

**Authors:** Aleksandra Miletić, Ivan Ristić, Maria-Beatrice Coltelli, Branka Pilić

**Affiliations:** 1Faculty of Technology Novi Sad, University of Novi Sad, 21000 Novi Sad, Serbia; alexm@uns.ac.rs (A.M.); ivan.ristic@uns.ac.rs (I.R.); 2Department of Civil and Industrial engineering, University of Pisa, 56122 Pisa, Italy; maria.beatrice.coltelli@unipi.it; 3National Inter University Consortium of Materials Science and Technology (INSTM), 50121 Florence, Italy

**Keywords:** coating, poly(lactide), chitin–lignin nanocomplex, grafting from, lactide oligomers

## Abstract

Recently, the demand for the use of natural polymers in the cosmetic, biomedical, and sanitary sectors has been increasing. In order to meet specific functional properties of the products, usually, the incorporation of the active component is required. One of the main problems is enabling compatibility between hydrophobic and hydrophilic surfaces. Therefore, surface modification is necessary. Poly(lactide) (PLA) is a natural polymer that has attracted a lot ofattention in recent years. It is bio-based, can be produced from carbohydrate sources like corn, and it is biodegradable. The main goal of this work was the functionalization of PLA, inserting antiseptic and anti-inflammatory nanostructured systems based on chitin nanofibrils–nanolignin complexes ready to be used in the biomedical, cosmetics, and sanitary sectors. The specific challenge of this investigation was to increase the interaction between the hydrophobic PLA matrix with hydrophilic chitin–lignin nanoparticle complexes. First, chemical modification via the “grafting from” method using lactide oligomers was performed. Then, active coatings with modified and unmodified chitin–lignin nanoparticle complexes were prepared and applied on extruded PLA-based sheets. The chemical, thermal, and mechanical characterization of prepared samples was carried out and the obtained results were discussed.

## 1. Introduction

Following the latest EU policies and strategies on plastics use, the implementation of bio-based materials is more than welcome in all industrial fields, especially for single-use products. The healthcare sector, including biomedical, cosmetics, and sanitary products, is one of the big consumers of single-use polymer-based products, and significant effort is put into changing fossil-based materials with bio-based materials [1]. Besides that, healthcare products need to fulfill many other requirements like safety, durability, efficacy, and biocompatibility. High-value products in this area have some functionalities, like anti-inflammatory, antimicrobial, and antioxidant functions, and represent an improved version of the previous ones [2]. Due to increased antimicrobial resistance problems, nowadays, natural active compounds are under investigation for use in the healthcare sector [3,4,5]. Thus, in the development of a new product for this purpose, researchers are facing two main issues: the right choice of bio-based material and suitable functionalization methods.

Biopolyesters, with poly(lactide) (PLA) as the most prominent representative, are an excellent alternative to fossil-based materials [6,7,8]. Sugar-based raw materials like corn and starchcan be used for the production of PLA, which makes it sustainable. PLA is biocompatible with the human organism and can be used in the production of implants as well as in other biomedical applications [9,10]. Also, it is biodegradable, with adjustable biodegradability according to the end-use requirements, and after degradation, only water and carbon-dioxide remains. It is certified as safe-to-use in healthcare products. PLA is a thermoplastic material, inert, with low interaction with cells, no functionality, and is resistant to acids, alkali, and fats. PLA is often used in drug-delivery systems, like carriers, for the controlled delivery of different medicines [11].

As evidenced in works regarding PLA bionanocomposites, one of the challenges in the development of functional films or medical devices is to achieve a good dispersion of active components within the polymer matrix [12,13] and, at the same time, maintaining their activity. Additionally, it is essential to ensure enough surfaceavailability of the component in order to achieve the desired activity [14,15]. Two main directions can be followed in the functionalization of materials: direct incorporation of active molecules into the material and physical or chemical bonding of actives onto the surface of the material. Since natural compounds are thermosensitive and deteriorate when exposed to high temperatures, direct incorporation is very challenging. Moreover, the difference in surface properties and hydrophilic–hydrophobic interactions makes successful incorporation even more difficult. One of the possible routes can be a surface modification of active compounds using some components with better compatibility with PLA, such as the ˝grafting from˝ method of modification [16]. The grafting of the polymer chain on a solid surface is a very adaptable method for surface modification and functionalization. Polymer chains can be grafted to the solid substrate (grafting to), or the grafting reaction can be proceeded by polymerization from the surface (grafting from). Both methods are suitable for forming a thin layer on the solid surface with the desired physical and chemical surface properties [17,18].

Bonding active molecules onto the surface often requires pretreatment of the surface for activation, as PLA surface is known as inert, and it is hard to attach something on it chemically. It is usually undertakenby treatment with aggressive chemicals [19,20], resulting in the introduction of new chemical groups like OH-, which are more reactive. This approach is suitable in low-volume products, where safety concerns are not so important. However, in medical products, theycan be a problem due to possible residues of chemicals that can induce irritations or allergies. Another possibility is plasma surfaceactivation, but this method is expensive for high-volume products, and the effect of the modification is generally modest [21,22,23].

A more straightforward method of surface activation, which is suitable for high-volume products, is cost-effective and still gives results, is coating with bio-based polymer active coating. It is favorable, but not limited to waterborne coatings, like polyurethane dispersions [24].

To obtain a very high adhesion between PLA-based films and the coating on it, the use of a PLA-based coating can be advantageous. PLA-based coatings were widely investigated because of their biocompatibility and biodegradability on several material surfaces.

For example, slow drug delivery systems based on biodegradable poly-lactic acid and antibiotic loaded hydroxyapatite microspheres were developed to be applied as coating on metal implants to prevent post-operative infections [25]. Chang et al. [26] prepared biodegradable corn starch films and used a PLA coating to improve theirwaterproof performance. Compared to the starch film without PLA coating, the films coated with PLA significantly reduced water solubility and increased the mechanical stability. Sputtering deposition and plasma assisted atomic layer deposition were also used for surface modification and the functionalization of PLA films, but those processes are not easily scalable to the industrial level [27,28].

This paper represents an attempt to select simple methodologies for modifying PLA-based film surfaces by including anti-microbial and anti-oxidant complexes based on chitin nanofibrils and lignin containing glycyrrhetic acid. In slightly acidic water suspension, chitin nanofibrils are positive, whereas the lignin is negative, hence they form a chitin–lignin (CN–LG) complex eventually entrapping selected functional molecules. This complex became cytocompatible, showing anti-inflammatory activity, and may serve for the delivery of biomolecules for skin care and regeneration [29]. In the present paper, the complex containing niacinamide is also considered. Niacinamide anti-inflammatory properties make it an attractive treatment for skin conditions marked by inflammations [30].

This work can be divided into three parts. First, surface modification of chosen active compounds was carried out by a “grafting from” method. Second, unmodified and modified active compounds were incorporated into solution cast PLA-based films, and properties were compared. Furthermore, in the third part, PLA-based coatings with unmodified and modified active compounds were prepared and applied to PLA-extruded film. The morphological, mechanical, thermal, and chemical properties were investigated.

## 2. Materials and Methods

### 2.1. Materials

High molecular poly(lactide) (PLA, Nature Works 2003 D, Minnetonka, MN, USA) and low molecular poly(lactide) (PLLA, Condensia Quimica, Barcelona, Spain, Mw 2000) were used as the polymer matrix. L-lactide (Sigma Aldrich, St. Louis, MI, USA) was used as a monomer for poly(lactide) synthesis, dichloromethane (Fisher Scientific, Hampton, NH, USA) as the solvent, and reaction medium for synthesis, H_2_SO_4_ acid (Sigma Aldrich, Taufkirchen, Germany) was used as the initiator of cationic polymerisation of lactide. Chitin–lignin nanoparticles loaded with niacinamide and glycyrrhetic acid were received from MAVI, produced in Texol, Alanno Scalo, Italy and were used as the initiator. They were produced according to patented technology [31]. All chemicals were used as received. The PLA-based extruded film was prepared according to a previous methodology by flat die extrusion [32]. The specific film was obtained by plasticizing with acetyl tri-n-butyl citrate (ATBC) a PLA/poly(butylene adipate co-terephathalate) (PBAT) blend [33].

### 2.2. Chitin–Lignin Nanoparticles Surface Modification

The chitin–lignin nanoparticle surface was modified by the “grafting from” method forming the PLA oligomers (OLA) at the particles surfaces by cationic polymerizationusing L-lactide as the monomer.

First, 1 g of chitin–lignin nanoparticles were dispersed in 60 mL of dichloromethane. Then, 10 g of the lactide monomer and 0.1 wt% (calculated on PLA weight) of a strong acid, sulfuric acid (H_2_SO_4_), as the initiator of the polymerization, were added. The reaction mixture was placed in a 250 mL balloon and connected to a reflux condenser. The synthesis was carried 5 h at 36 °C until all the particles become soluble in dichloromethane, which confirmed that all the particles were grafted by PLA oligomer. The grafted chitin–lignin powder was obtained after evaporation of solvent and drying at 40 °C for 10 h.

### 2.3. Preparation of PLA-Based Films by Solution Casting Method

PLA-based films (based on PLA 2003 D) with unmodified and surface-modified Chitin–lignin complex nanoparticles were prepared by the solution casting method. First, 10 wt% solution of PLA in dichloromethane was prepared, and then, in the second step, 1 wt% (calculated on PLA weight) of nanoparticles was added. After mixing at room temperature, the dispersion was poured out in Petri dish, and films were obtained after drying 24 h on the air in the room temperature.

### 2.4. Preparation of Poly(L-Lactide) (PLLA)-Based Coating

Coatings containing unmodified and surface-modified Chitin–lignin complex nanoparticles were prepared using low molecular weight PLA (PLLA) by dissolving it in dichloromethane in the concentration of 30 wt% and adding 1 wt% (calculated on PLLA weight) of appropriate chitin–lignin complex. The prepared coating was applied to PLA-based extruded film using a brushing technique.

### 2.5. Characterization Methods

The light microscope was used for onsite examination of sample morphology.

Chemical properties of samples were examined using Attenuated Total Reflectance-Fourier Transform Infrared spectroscopy (ATR-FTIR) Shimadzu IRaffinity equipment, Kyoto, Japan, by scanning samples from 4000 to 400 cm^−1^, with a resolution of 4 cm^−1^, and following changes in obtained spectra.

Thermal properties were determined using the differential scanning calorimetry (DSC) method on TA Instruments Q20 equipment (New Castle, DE, USA) by heating samples with a heat flow of 10 °C/min in one cycle. The samples were sampled from the casted films and from the coated film made of plasticized PLA/PBAT including all the film thickness.

Shimadzu EZ-Test, Kyoto, Japan instrument was used for mechanical properties assessment. Samples were cut in a rectangular shape of dimensions 1 cm × 4 cm, 0.3 mm thickness and tested with a clamp speed of 10 mm/min.

## 3. Results and Discussion

The cationic solution polymerization was selected for the grafting of lactide oligomers on the chitin–lignin–glycyrrhetic acid complex because it is possible to perform it even at a temperature of 40 °C, yielding PLA with narrow molecular weight distribution in a very short period of time (less than 6 h), on the basis of our previously work [34]. It was shown that a different macroinitiator could be used to promote lactide polymerization enabling desired control of the properties of the final product [35].

The grafting was performed onto two different CN–LG systems: one containing glycyrrhetic acid [36] and the second with niacinamide (Figure 1). These two different trials were performed to consider the reliability of the polymerization despite of the different chemical structure of the functional molecule.

In both cases, grafting was successful, and the chemical structure of grafted CN–LG complexes was confirmed by FTIR analysis (Figure 2a,b). In fact, in both PLA grafted CN–LG complexes, characteristic bands for the complex are present with additional bands for PLA (such as C=O carbonyl at 1760 cm^−1^ and 1090 cm^−1^ for C–O–C stretching).

From the DSC results of unmodified and grafted chitin–lignin complexes (Figure 3a,b), it can be concluded that the grafting of PLA onto complex surfaces has a strong influence on their thermal properties. While chitin–lignin complexes show the typical peak of chitin attributed to water loss [37,38] at a temperature at about 50 °C and glass transition of lignin around 150 °C [39], modified complexes grafted with PLA showed thermal properties similar to PLA, with slightly lower Tg at around 26 °C, a crystallization temperature (Tc) value around 80 °C with broad cold-crystallization peak (Tcc), and melting temperature (Tm) around 140 °C. This can indicate that the PLA layer formed around the chitin–lignin nanoparticle protects it from the direct influence of high temperatures and can maintain its activity.

Unmodified and modified active compounds were incorporated into solution-cast PLA-based films, and the properties were compared. The PLA used, in this case, is the general grade PLA used for extrusion and various applications. It can be seen that, according to the color of prepared films (Figure 4), the applied grafting method improved the compatibility of PLA and the selected complex. Films with grafted complexes are transparent, similar to the pure PLA film, which confirmed that grafted complexes were well dispersed in the PLA matrix.

Unmodified complex loaded films have intense colors of the complex, which indicated phase separation due to the low compatibility of the PLA and complex. This is confirmed by optical microscopy (Figure 5a,b), where complex aggregates are visible as dark regions. PLA films with grafted chitin–lignin complexes did not show any aggregate, and figures are transparent as pure PLA films.

Changes in band position and intensity in FTIR spectra of PLA films with unmodified and PLA grafted chitin–lignin complexes (Figure 6 and Figure 7) were not detected. These results confirmed that in the grafting method, complexes were not entirely covered by PLA, but in such a way that compatibility was improved without the loss of functionality.

DSC thermograms of PLA films with unmodified chitin–lignin complexes, (Figure 7a and Figure 8b), showed the characteristic transition temperatures of PLA; a glass transition temperature at 57 °C and 40 °C, Tc at 116 °C and 94 °C and Tm at 150 °C for film with niacinamide and glycyrrhetic acid, respectively.

The significant differences in the recorded values can be attributed to the different behavior of glycyrrhetic acid and niacinamide. It is possible that niacinamide is responsible forthe formation of hydrogen bonds with PLA and CN–LG, because of its amide groups and electron rich nitrogen in the pyridine ring. This can explain the higher value of Tg, due to reduced chain mobility, and the higher value of cold crystallization temperature, due to the increased disorder and the energy barrier in the material. Moreover, the sample films with unmodified chitin–lignin and glycyrrhetic acid showed a higher crystallinity than the one with niacinamide. Glycyrrhetic acid, having a higher molecular weight and thus a high molecular dimension [33], and a carboxylic function [39], is probably better in inducing heterogeneous nucleation in PLA [40]. In fact, its 3D structure consists of a planar system with a methyl group in opposition, almost perpendicular to the plan. As PLA has CH_3_ groups on its chain, the capacity of the glycyrrhetic acid to interact with PLA, inducing its local organization in crystals, is thus significant, and seems similar to the nucleating action due to the formation of PLA stereocomplexes [41].

Chitin–lignin complex grafted with PLA had a strong influence on PLA properties: in the first cycle of the heating, the peak of cold crystallization disappeared, the Tg value decreased by 12 °C, and melting temperature increased up to 156 °C (Figure 7c), typical of the crystal form [42]. These results showed that the presence of the CN–LG-niacinamide-g-PLA induced the crystallization of PLA during the cooling occurred before the studied heating. Hence the CN–LG-niacinamide-g-PLA behaved as a nucleant. The crystallinity of the film is thus very high, and for this reason the cold crystallization did not occur. Interestingly, the high value of melting point can be ascribed to the presence of more regular crystal than in the case of unmodified chitin–lignin complexes. It is reasonable that the PLA chains grafted onto the CN–LG complex co-crystallize with PLA, thus better promoting the development of crystals. PLA block copolymers have shown a similar behavior, inducing an easier crystallization of PLA blocks [43]. In good agreement, the addition of silica-grafted-PLA to PLLA also induced a strong increase in crystallinity due to this nucleating effect [44]. All the improvements can be ascribed to stronger entanglements between PLLA stretched by nano-SiO_2_ and PLA matrix.

The incorporation of both modified and unmodified chitin–lignin complexes nanoparticles influenced the mechanical properties of PLA. In both cases, mechanical properties were improved. When unmodified chitin–lignin nanoparticles loaded with active compounds were added to PLA, the tensile strength was increased almost ten times, together with the improvement of elasticity for three to four times. Better elasticity was achieved by the incorporation of chitin–lignin complex nanoparticles modified with PLA due to the better dispersion of particles within the PLA matrix, which is a direct consequence of better compatibility of these particles with PLA. Compared to the mechanical properties of PLA, films with modified particles had around four times better tensile strength and six times better elasticity. However, if the mechanical properties of the same samples were compared to ones with unmodified particles, it can be observed that tensile strength values were slightly lower. Nevertheless, the elongation at break was improved—it almost doubled. These results are in accordance with the optical and thermal properties of the samples. The results of mechanical testing are summarized in Table 1.

The obtained results showed that CN–LG complexes can be incorporated, grafted or ungrafted, into PLA films by solution casting, and they can allow to strongly affect and modulate mechanical properties of these films.

Nevertheless, the possible application of poly(lactide)-based films as coating onto an extruded film is veryinteresting. For this activity, a specific oligomeric PLA witha molecular weight of 2000 was used (PLLA). This polymer was selected because it can be potentially used in the future to prepare coatings in common solvents, thanks to its lower molecular weight and thus enhanced solubility.

After this, extruded PLA films were coated with PLLA-based coating containing unmodified and surface modified chitin–lignin complex nanoparticles. As in the previous case, the difference in the color of the two films can be seen. While PLLA loaded with unmodified particles has a brownish color (Figure 8a), originating from the color of the complex, coating with modified particles has a lighter color (Figure 8b).

Figure 9a,b illustrate the appearance of the coating with unmodified particles by an optical microscope. It can be seen that particles are mostly in the form of aggregates and that the distribution of particles is extensive, from very small to massive aggregates. By grafting PLA on the surface of chitin–lignin complexes, this aggregation of particles is avoided, and better dispersion was achieved. This is important for the activity of the coating, because in the case where aggregates are present, the number of domains dispending active functionalities is lower than in the case of single particles present.

Figure 10 illustrates the FTIR spectra of uncoated and coated PLA extruded films, with unmodified complexes loaded coating. The PLLA-based coating did not induce a change in the infrared spectrum of the plasticized PLA/PBAT extruded film. The same result was obtained when modified complexes were added to the coating. On the other hand, the PLLA spectrum is similar to the one of the film and it is the main component of the coating.

Comparing DSC thermograms, related to the first heating of uncoated and coated PLA films (Figure 11a,b), with PLLA-based coating containing unmodified chitin–lignin complexes, significant differences in Tc and Tm can be detected. While the crystallization peak of neat plasticized PLA/PBAT film is regularly shaped, the coated film has a broad crystallization peak, with crystallization occurring at a lower temperature, which indicates the changes in the crystallization behavior of PLA where two peaks, crystallization of the oligomer and crystallization of the film, are merged. On the other hand, the melting point is also lower for the coated films. Neat PLA film has two melting points nicely separated, which is typical for PLA derived from α and α’ crystal forms. Within coated PLA, the intensity of the ordered α form decreased, which might indicate that the added coating, rapidly forming a film by solvent evaporation, is contributing with a more disordered α’ crystal form. However, all the observed effects are the result of cumulative phase transitions within different PLA-based layers. In fact, we cannot exclude some surficial dissolution or swelling due to the solvent on the surficial layer of the plasticized PLA/PBAT film.

Similar results were obtained when coatings with modified chitin–lignin complexes were applied, as it can be seen from the results summarized in Table 2. The Tg is slightly lower compared to other PLA films, which might be the effect of additional PLA layer grafted on chitin–lignin complexes, adding flexibility and disorder to the coating layer. Moreover, the deposition of the thin coating film containing solvent can slightly affect the film crystallinity.

The mechanical properties of uncoated and coated PLA films are summarized in Table 3. Values of tensile strength and elongation at break for uncoated and coated PLA films are almost the same, which indicates that the coating did not affect the mechanical properties of neat PLA. The application of the coating on a preformed extruded film is a good methodology to keep the nano-structured functional molecules on the film surface without modifying its properties.

## 4. Conclusions

The modification of chitin–lignin complexes with PLA was successfully carried out by employing the “grafting from” method, grafting PLA chains onto chitin–lignin complexes containing glycyrrhetic acid or niacinamide. These grafted systems were dispersed in an extrusion grade PLA or in an oligomeric PLA (PLLA). The polylactide grafted chains enable a better dispersion and compatibility with PLA. The physical modification of PLA, which was confirmed by FTIR, was undertaken by the direct incorporation of unmodified and modified chitin–lignin complexes into solution casting PLA-based films, and by coating based on low-molecular weight PLLA containing mentioned active compounds. When particles were directly incorporated into the films, the effect on mechanical and thermal properties was evident. PLA-based films with modified particles incorporated had lower Tg and higher Tm values, while they had no cold crystallization peak. In fact, these films showed an increased crystallinity, thanks to the nucleation action of the nanostructured additives. Due to the better dispersion, films with modified particles had better elasticity compared to neat and film containing unmodified particles. When coating was applied on the surface of the extruded film based on PLA, the effect on mechanical and thermal properties was lower, and the uncoated and coated film had similar properties, no matter which active compound they contained. Depending on the desired application, value-to-cost ratio, price of the product, and volume of production, it is possible to exploit the thermo-mechanical or functional (anti-microbial and anti-inflammatory) properties of chitin–lignin complexes, incorporating or depositing them onbiodegradable films. These preliminary studies yielded these findingsand opened several routes for PLA-based films intheir application on contact with skin.

## Figures and Tables

**Figure 1 jfb-11-00030-f001:**
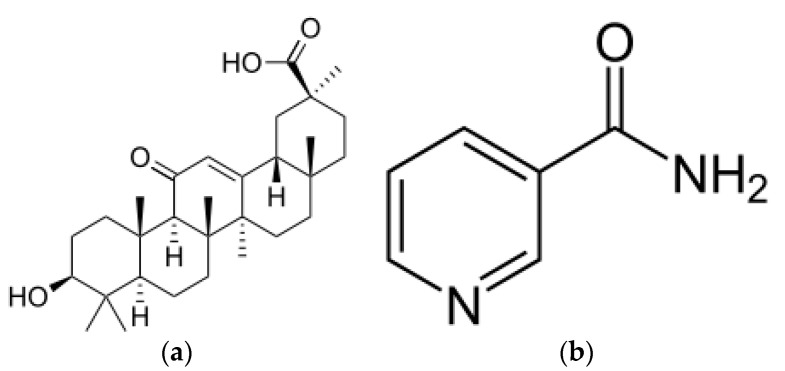
Molecular structure of (**a**) glycyrrhetic acid; (**b**) niacinamide.

**Figure 2 jfb-11-00030-f002:**
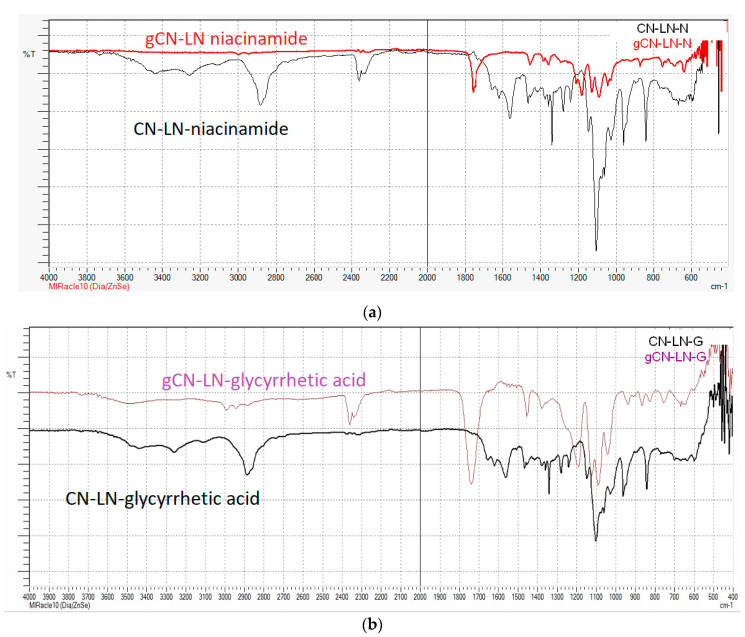
FTIR spectrum of: (**a**) Poly(lactide) (PLA) grafted chitin–lignin complex with niacinamide; (**b**) PLA grafted chitin–lignin complex with glycyrrhetic acid.

**Figure 3 jfb-11-00030-f003:**
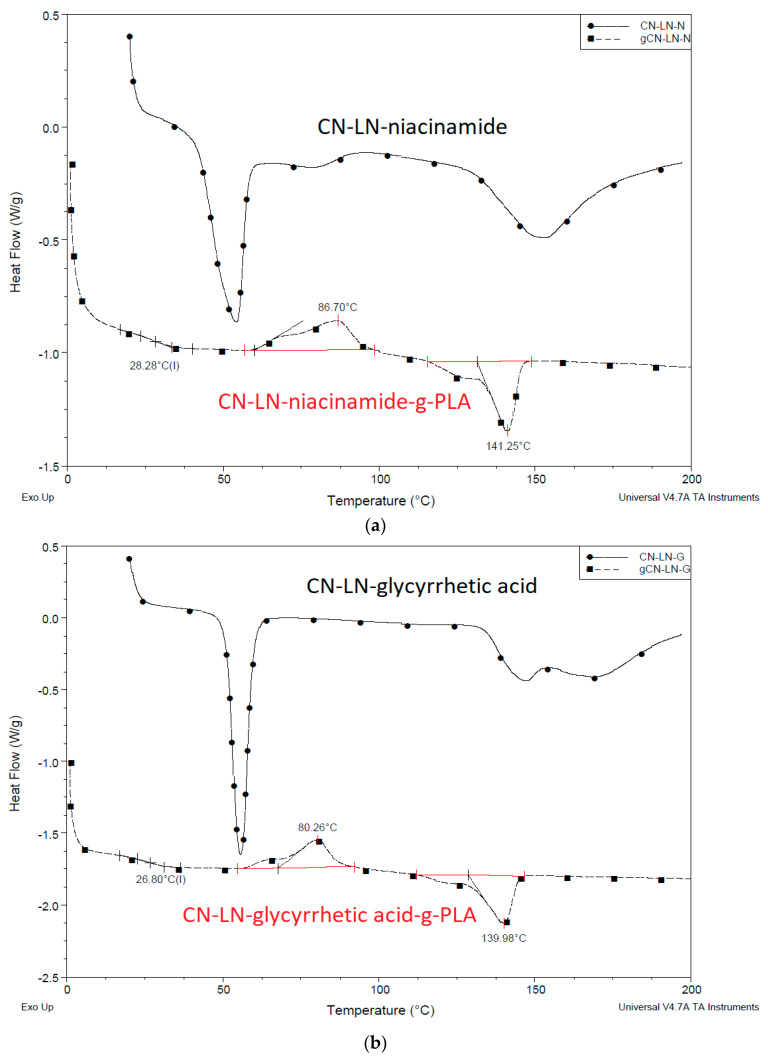
(**a**) Differential scanning calorimetry (DSC) thermogram of unmodified and modified chitin–lignin complexes by PLA grafting niacinamide complexes (**b**) glycyrrhetic acid.

**Figure 4 jfb-11-00030-f004:**
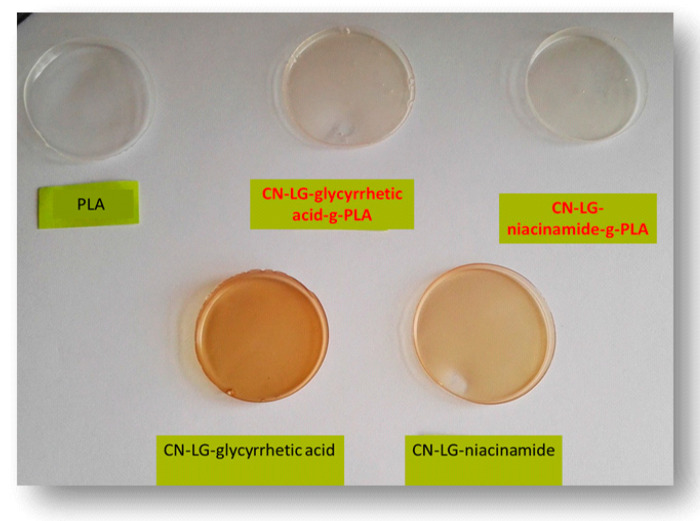
PLA-based films obtained via a solution casting method; first raw left to right PLA, PLA with chitin–lignin complex with glycyrrhetic acid modified by grafting, PLA with chitin–lignin complex with niacinamide modified by grafting, second raw PLA with unmodified chitin–lignin complex with glycyrrheticacid and PLA with unmodified chitin–lignin complex with niacinamide.

**Figure 5 jfb-11-00030-f005:**
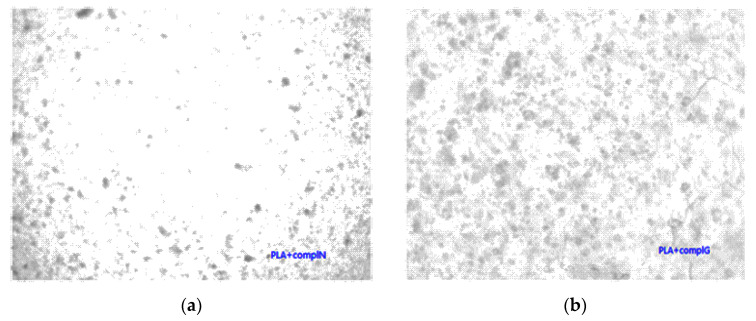
Micrographs (magnification 100×) of solution cast films of PLA with added (**a**) chitin–lignin complex with niacinamide and (**b**) chitin–lignin complex with glycyrrhetic acid.

**Figure 6 jfb-11-00030-f006:**
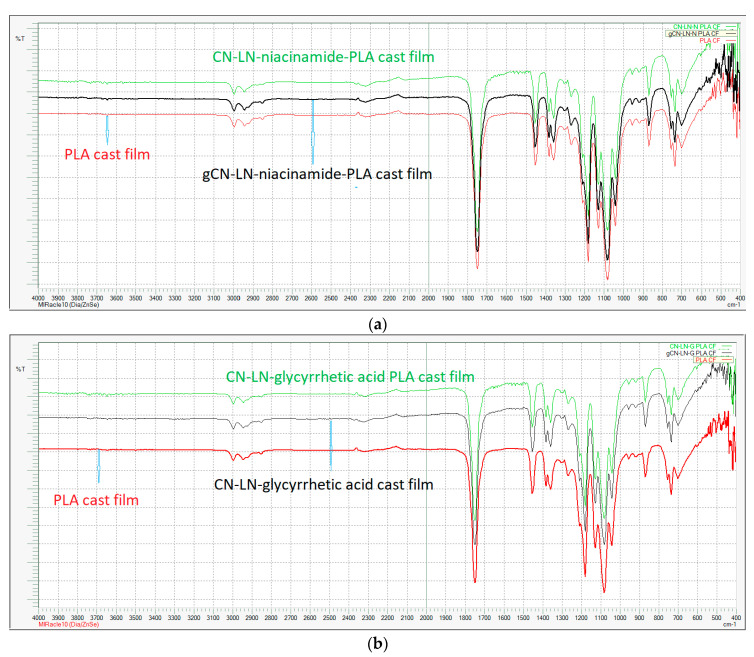
FTIR spectra of PLA cast films with (**a**) unmodified chitin–lignin complexes and (**b**) modified chitin–lignin complexes.

**Figure 7 jfb-11-00030-f007:**
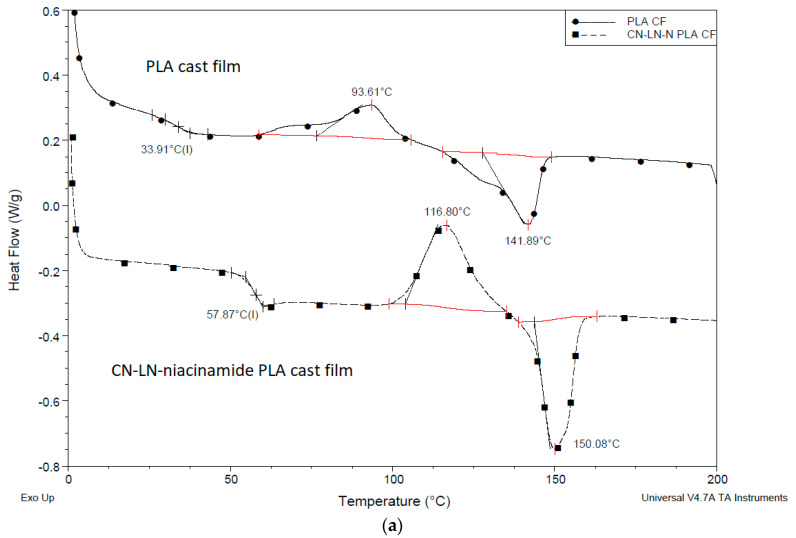
DSC thermogram of PLA film with chitin–lignin complex (**a**) unmodified with niacinamide, (**b**) unmodified with glycyrrhetic acid, (**c**) grafted chitin–lignin complex with niacinamide.

**Figure 8 jfb-11-00030-f008:**
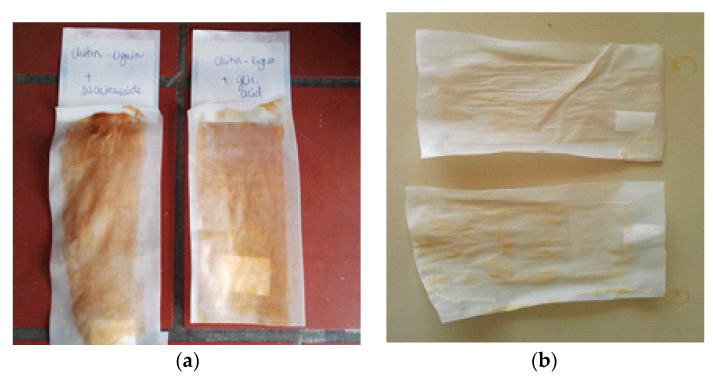
PLA extruded sheet coated with low molecular poly(lactide) (PLLA)-based coating with unmodified chitin–lignin niacinamidecomplexes (left) and modified chitin–lignin glycyrrhetic complexes (right), (**a**) and (**b**) modified chitin–ligninglycyrrhetic complexes (up) and modified chitin–lignin niacinamide complexes (down).

**Figure 9 jfb-11-00030-f009:**
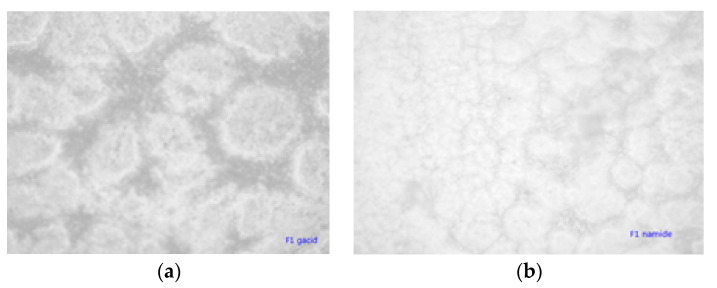
Micrographs (magnification 100×) of extruded film coated with PLLA-based coating with unmodified chitin–lignin complex loaded with, (**a**) glycyrrhetic acid (**b**) niacinamide.

**Figure 10 jfb-11-00030-f010:**
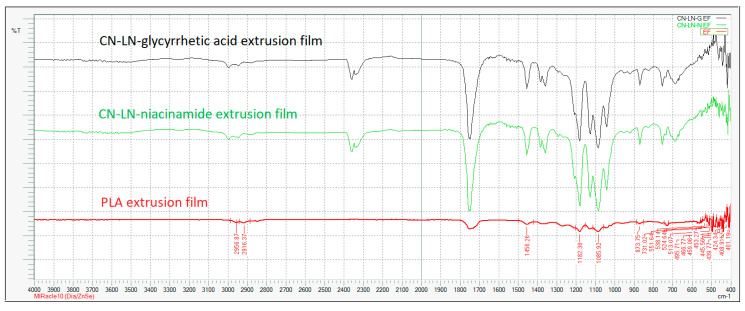
FTIR spectra of PLA extruded film coated unmodified chitin–lignin complex nanoparticles.

**Figure 11 jfb-11-00030-f011:**
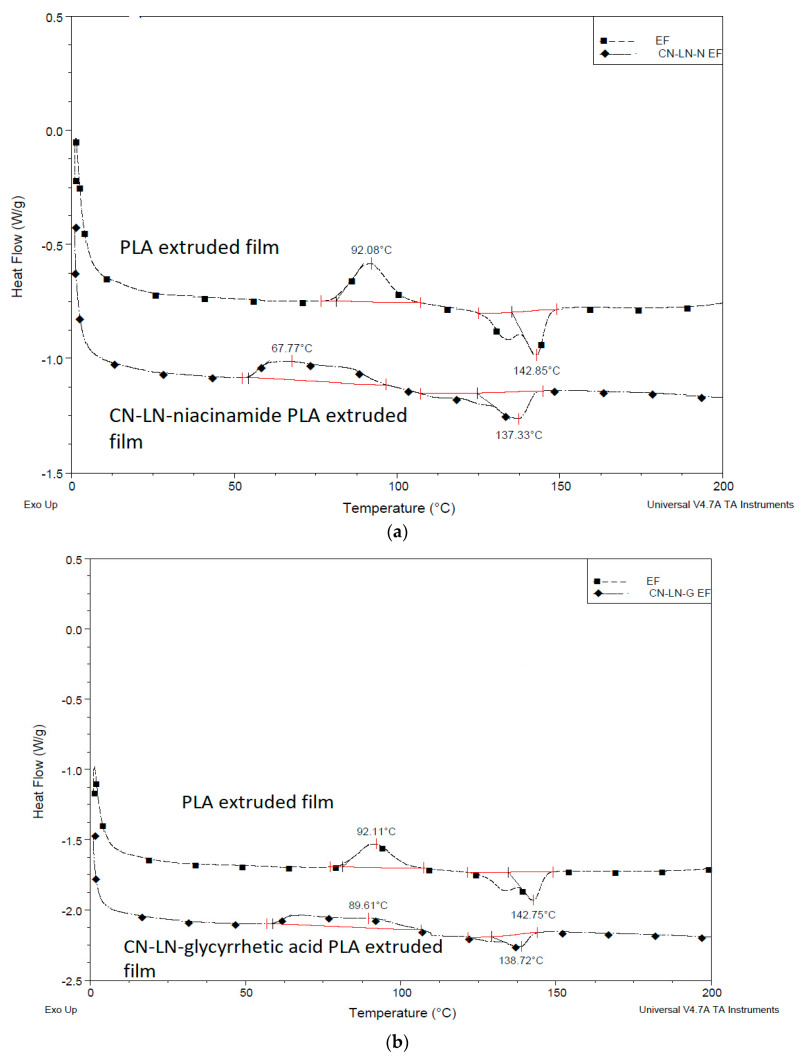
DSC thermogram extruded film coated with PLLA-based coating with (**a**) unmodified chitin–lignin niacinamide complexes and (**b**) unmodified chitin–lignin glycyrrhetic acid complexes.

**Table 1 jfb-11-00030-t001:** Mechanical properties solution casting film.

Sample	Tensile Strength (MPa)	Elongation at Break (%)
PLA cast film	1.96	44.33
CN-LN-niacinamide PLA cast film	11.75	155.13
CN-LN-glycyrrhetic acid PLA cast film	11.61	112.82
gCN-LN-niacinamide PLA cast film	9.24	264.66
gCN-LN-glycyrrhetic acid PLA cast film	7.92	236.78

**Table 2 jfb-11-00030-t002:** Thermal property results of the extruded film (EF) coated by PLLA-based coating.

Sample	Tg (°C)	Tc (°C)	Tm (°C)
PLA extruded film	46.7	92.14	142.8
CN-LN-niacinamide PLA extruded film	45.2	83.58	137.5
CN-LN-glycyrrhetic acid PLA extruded film	47.6	89.88	138.72
gCN-LN-niacinamide PLA extruded film	44.3	84.7	136.2
gCN-LN-glycyrrhetic acid PLA extruded film	43.1	85.2	136.8

**Table 3 jfb-11-00030-t003:** Mechanical results properties of the EF coated by PLLA-based coating.

Sample	Tensile strength (MPa)	Elongation at break (%)
PLA extruded film	0.3	112
CN-LN-niacinamide PLA extruded film	0.26	115.2
gCN-LN-niacinamide PLA extruded film	0.25	116.54
CN-LN-glycyrrhetic acid PLA extruded film	0.24	114.76
gCN-LN-glycyrrhetic acid PLA extruded film	0.24	112.48

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
