# Peer review of "Modification of PLA-Based Films by Grafting or Coating"

_jfb, 2020, doi:10.3390/jfb11020030_

Round 1

Reviewer 1 Report

The manuscript entitled "Modification of PLA based films by grafting or coating" investigated the functionalization of PLA inserting antiseptic and anti-inflammatory nanostructured systems based on chitin nanofibrils-nanolignin complexes. The experiments appear to be well planned, results are interesting and correct, the ideas and methods are correct. In my mind, the manuscript is acceptable for publication in Journal of Functional Biomaterials after minor revision. Some suggestions were listed below: (1) researchers are facing two main issues: the right choice of bio-based material and suitable functionalization methods [6]. 6 Jiawei Yan, Zisheng Luo, Zhaojun Ban, Hongyan Lu, Dong Li, Dongmei Yang, Morteza Soleimani Aghdam, Li Li. The effect of the layer-by-layer (LBL) edible coating on strawberry quality and metabolites during storage. Postharvest Biology and Technology, 147(2019): 29-38. The quality of Figure 5 is too low.

Author Response

Response to Reviewer 1 Comments

29-38. The quality of Figure 5 is too low.

The quality of image 5 has been improved

Reviewer 2 Report

A work of good scientific level with minor changes to be made! ​A good synthesis of the literature offering an overview of the evolution researches in the area.

This manuscript, entitled „Modification of PLA based films by grafting or coating” is considered to be relevant to the scope of this journal.

However, several points need to be addressed prior to publication of this manuscript. My comments/suggestions are given:

  1. Figure 5 must be replaced because it is not clear.
  2. The scale must be inserted in Figure 9.
  3. Authors should check the references input mode.
  4. I think it would be useful to study the wettability to establish hydrophilic or hydrophobic character of the new obtained materials.

Author Response

Response to Reviewer 2 Comments

Point 1. Figure 5 must be replaced because it is not clear.

Response 1:  1 Figure 5 has been replaced

Point 2: The scale must be inserted in Figure 9.

Response 2:  Micrographs (magnification 100 x)  of extruded film coated with PLLA-based coating with unmodified Chitin-Lignin complex loaded with, (a) glycyrrethic acid (b) niacinamide

Point 3: Authors should check the references input mode

Response 3: References input mode has been checked and corrected.

Point 4:  I think it would be useful to study the wettability to establish hydrophilic or hydrophobic character of the new obtained materials.

Response 4: Yes, wettability will be useful.  However in this case, we consider that measuring of wetting angle is not essential. It should not been changed, because PLA is grafted at the CN-LN particles surfaces, and they should have the same surface tension, hydrophobicity / hydrophilicity as pure PLA film.

Reviewer 3 Report

This paper focused on the modification of PLA surface with grafting or coating. 

  1. Too much information if the introduction part, please only introduce the related background and less repeat information;
  2. The words Poly(lactide)(PLA), poly(lactic acid)(PLA), polylactic acid(PLA), I found many different spell in the manuscript. Please check all the manuscript and edit carefully;
  3. Only FT-IR was used to confirm the success of the grafting, which is not enough and convinced, NMR and other methods should be performed; and how much were grafted on the surface? 
  4. Figure 4 just showed the color of the films, I could not understand the meaning of that. 
  5. Again, all data should be well organized to support your conclusion. 

Author Response

Response to Reviewer 3 Comments

Point 1: Too much information if the Introduction part, please only introduce the related background and less repeat information;

Response 1: In paragraph 2 of the Introduction part, the last sentence has been omitted: 

"Nevertheless, there are some drawbacks in the mechanical and thermal properties of PLA, which can be overcome with the addition of appropriate additives."

Paragraphs 8  and 9 of the Introduction part have been omitted.

"The electrospinning technique is often considered for surface modification of PLA films. Karacan et al. [29] produced bi- and multilayer poly(lactide (PLA)films by the incorporation of electrospun nanostructured PLA coatings and interlayers containingthe antioxidant gallic acid (GA) at 40 wt% onto cast-extruded PLA films. The modified PLA films can perform as potent vehicles to deliver GA.

Polylactic acid (PLA) films were coated by coaxial electrospinning with essential and vegetable oils (clove and argan oils) and encapsulated into chitosan, to combine the mechanical properties and biodegradability of PLA substrates with the antioxidant and antimicrobial properties of the chitosan–oil nanocoatings [30]. Interestingly, chitin nanofibrils, obtained by chitin, abundantly present in sea food waste or mushrooms showed cells regenerative and anti-microbial properties similar to chitosan [31, 32] and were used in tissue engineering and cosmetics [33, 34]."

Point 2: The words Poly(lactide)(PLA), poly(lactic acid)(PLA), polylactic acid(PLA), I found many different spell in the manuscript. Please check all the manuscript and edit carefully;

Response 2: The word Poly(lactide) (PLA) has been chosen.

Point 3: Only FT-IR was used to confirm the success of the grafting, which is not enough and convinced, NMR and other methods should be performed, and how much were grafted on the surface? 

Response 3: At the beginning of the grafting reaction, only the lactide monomer is dissolved in dichloromethane, and CN-LN particles are in a forme of a dispersion.

As the reaction is performing, CN-LN particles become soluble because PLA chains are grafted on the surfaces of the particles. The solubility of CN-LN particles in dichloromethane is the confirmation that grafting of PLA on the surfaces of the particles occurred. The pure CN-LN particles were not present in the reaction mixture, at the end of the reaction.

Regarding the amount of the grafting PLA on the surfaces of the particles, the ratio of lactide monomer/CN-LN complexes is defined according to the amount of the active CN-LN particles in the final material, not by the amount of grafted PLA. Therefore, the monomer addition was in excess to obtain the coverage of the whole amount of CN-LN particles in the reaction mixture.

The aim of this work was to make compatible CN-LN particles with PLA; therefore, we didn't perform additional testing as NMR. For example, NMR will confirm the polymerization of PLA and chemical bond between PLA chains and CN-LN particles surfaces, which was approved by the solubility of CN-LN particles.

Point 4: Figure 4 just showed the color of the films. I could not understand the meaning of that. 

Response 4: Figure 4. PLA-based films obtained via a solution casting method; first raw left to right PLA, PLA with chitin-lignin complex with glycyrrethic acid modified by grafting, PLA with chitin-lignin complex with niacinamide modified by grafting, second raw PLA with unmodified chitin-lignin complex with glycyrrethic acid and PLA with unmodified chitin-lignin complex with niacinamide 

The colorlessness of the films in the first row confirms better solubility and compatibility of the modified CN-LN particles by grafting with PLA.

The color of the virgin CN-LN particles in the form of the powder is brown.

Unmodified complex loaded films have the intense color of complex, which indicated phase separation due to the low compatibility of PLA and complex. It is confirmed by optical microscopy (Figure 5a and 5b), were complex aggregates are visible as dark regions. PLA films with grafted Chitin-Lignin complexes did not show any aggregate, and figures are transparent as pure PLA films.

Point 5: Again, all data should be well organized to support your conclusion. 

Response 5: The answer is given in response 3.

Round 2

Reviewer 3 Report

agree